# Electromagnetic Characteristic Analysis of Permanent Magnet Synchronous Machine Considering Current Waveform According to Static Rotor Eccentricity

**Hoon Ki Lee, Tae Kyoung Bang , Jong Hyeon Woo, Hyo Seob Shin and Jang Young Choi ***

Department of Electrical Engineering, Chungnam National University, 99 Daehak-ro, Yuseong-gu, Daejeon 34134, Korea; lhk1109@cnu.ac.kr (H.K.L.); bangtk77@cnu.ac.kr (T.K.B.); dnwhd0@cnu.ac.kr (J.H.W.); shs1027@cnu.ac.kr (H.S.S.)

*** Correspondence: choi_jy@cnu.ac.kr; Tel.: +82-042-821-7601

**Abstract:** In this study, we performed an electromagnetic characteristic analysis of a permanent magnet synchronous machine considering the current waveform based on static rotor eccentricity. First, the characteristics of the back electromotive force were analyzed through the no-load analysis of the analysis model according to static rotor eccentricity. Next, a dynamic analysis was performed through mathematical modeling of a permanent magnet synchronous motor and PWM method. The input current during operation was derived through the dynamic analysis, and the core loss analysis was performed using derived input current. The core loss analysis was performed using the case where the fundamental wave current was applied and the input current derived through the dynamic analysis, and the results were compared.

**Keywords:** PMSM; dynamic analysis; eccentricity

## 1. Introduction

Permanent magnet synchronous machines (PMSMs) are becoming essential for applications such as home appliances, industrial tools, and electrical vehicles. Accordingly, interest in motor faults is increasing, and remedying motor faults at industrial sites is crucial. The statistics provided in a recent study [1] show that 41% of motor faults are bearing faults, 37% are stator faults, 10% are rotor faults, and 12% are miscellaneous faults. Therefore, motor malfunction is being actively investigated. One of the factors contributing to motor malfunction is rotor eccentricity, wherein the center of the rotor axis deviates from the center of the stator, rendering the air gap non-uniform. Rotor eccentricity interrupts the uniform distribution of the magnetic flux to the stator, resulting in an increased cogging torque and an unbalanced magnetic force. This adversely affects the motor performance [2–5]. In the case of severe eccentricity, the air-gap size is reduced significantly, and an accident occurs between the stator and rotor. Consequently, the insulation between the stator core and stator winding may be destroyed, which may cause a fire in the electric motor. It is important to prevent this in advance as eccentricity occurs in the rotor. Methods for diagnosing eccentricity include the TIR test using a dial indicator, impedance measurement using an LCR meter, and impedance measurement after inputting a diagnostic signal using an inverter. As described above, studies regarding eccentricity diagnosis have been actively conducted. In a previous study, the analysis of current harmonics generated when eccentricity occurred was performed via a dynamic analysis [6–9]; however, the characteristics of the motor owing to the current harmonics were not analyzed. To analyze the dynamic characteristics of a motor, the whole drive system must be modeled to confirm the current, voltage, and torque characteristics via a coupled analysis; however, this is time consuming. Hence, a co-simulation was

devised. In a co-simulation, the data are primarily stored based on the rotor position. The results are the same as those of coupled analysis, but require comparatively less process time.

In this study, we performed an electromagnetic characteristic analysis of a PMSM considering the current waveform based on rotor eccentricity. First, the characteristics of the back electromotive force (EMF) affecting the control aspect were analyzed based on the rotor eccentricity. Next, we confirmed the harmonics of the input current based on the rotor eccentricity through a dynamic analysis. The tendency of the generated current harmonics was confirmed by comparing the experimental results. Finally, we compared the considered and non-considered current harmonics.

## 2. PMSM Control

To analyze the dynamic characteristics of the PMSM due to rotor eccentricity, the drive system was modeled mathematically using the voltage equation of the PMSM; subsequently, the selected PWM method, inverter switch, and controller were modeled.

### 2.1. Analysis Model

Figure 1a shows the analysis model. Figure 1b,c shows the manufactured model. Table 1 lists the design specifications. An eight-pole, nine-slot PMSM was selected to confirm the effect on eccentricity. Steel 50PN470 was used as the core of the stator and the rotor, whereas N42SH was used as the magnet.

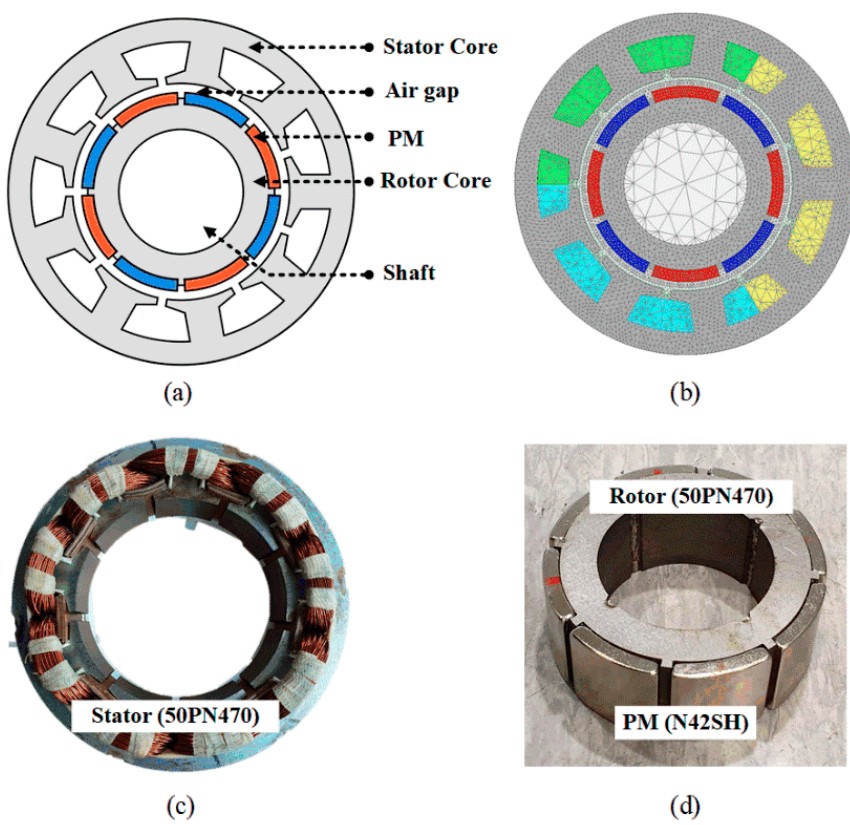

**Figure 1.** Analysis model: (**a**) eight-pole nine-slot permanent magnet synchronous machine (PMSM); (**b**) the mesh plot of analysis model; (**c**) manufactured stator; and (**d**) manufactured rotor.

**Table 1.** Design specifications.

| Parameter | Value | Unit |
|---|---|---|
| Number of slots/poles | 9/8 | |
| Outer radius of stator | 47 | mm |
| Outer radius of rotor | 43 | mm |
| Thickness of PM | 5 | mm |
| Axial length | 30 | mm |
| Magnet remanence | 1.27 | T |
| Pole arc ratio | 0.9 | |
| Rated speed | 1000 | rpm |
| Rated torque | 2 | N |
| Rated power | 200 | W |
| PWM method | SPWM | |

*2.2. Mathematical Modelling of PMSM*

The electrical dynamic equations were transformed from a three-phase rotating time frame (a, b, c) into a two-coordinate rotating position frame ($d$, $q$) by the Clarke and Park transformation [10]. Hence, the $d$-$q$ axis model of the three-phase dynamic equation under the steady state is expressed as

$$\begin{bmatrix} v_d \\ v_q \end{bmatrix} = \begin{bmatrix} R_s + pL_s & -wL_s \\ wL_s & R_s + pL_s \end{bmatrix} \begin{bmatrix} i_d \\ i_q \end{bmatrix} + \begin{bmatrix} 0 \\ w\lambda_{pm} \end{bmatrix} \tag{1}$$

where $v_d$ and $v_q$ are the voltages in the $d$-$q$ axis, and $w$ is the angular frequency. $R_s$, $L_s$, and $p$ are the phase resistance, synchronous inductance, and differential operator, respectively. The rotational speed of the motor can only be calculated by assuming that the torque point is in a steady state. The expansion of Equation (1) shows that the current and voltage are restricted by the supply voltage; therefore, the rotational speed can be expressed using the following equation [11]:

$$\left(\frac{V_{smax}}{w_r}\right)^2 \geq (L_d i_d + \lambda_{pm})^2 + (L_q i_q)^2 \tag{2}$$

$$w_r = \frac{V_{smax}}{\sqrt{(L_d i_d + \lambda_{pm})^2 + (L_q i_q)^2}} \tag{3}$$

where $V_{smax}$, $L_d$, $L_q$, $\lambda_{pm}$, $w_r$ are the voltage limit, $d$- and $q$- inductances, no-load flux, and base speed, respectively. To obtain speeds above the base speed, flux-weakening control was performed in the permanent magnet (PM) machines. The optimized current references for the torque reference in the flux-weakening region are as follows [12]:

$$i_d^* = \frac{L_d \lambda_f - \sqrt{(L_d \lambda_f)^2 + (L_q^2 - L_d^2)\left(\lambda_f^2 + (L_q I_s) - \left(\frac{V_{smax}}{w_r}\right)^2\right)}}{L_q^2 - L_d^2} \tag{4}$$

$$i_q^* = \pm\sqrt{I_{smax}^2 - i_d^{*2}} \tag{5}$$

where $\lambda_f$, $i_d^*$, and $i_q^*$ are the load magnetic flux and $d$- and $q$- current references, respectively.

*2.3. Vector Control of PMSM*

Figure 2 shows a block diagram for the dynamic simulation of the PMSM. As shown in the figure, the overall system includes two closed loops, an inner current control loop and an outer speed control loop.

Whenever a reference speed is specified, the system automatically compares it with the actual speed. The speed error directly affects the load torque profile based on the motor motion equation as follows [12]:

$$T_e = J_r \frac{dw_r}{dt} + B_f w_r + T_d \qquad (6)$$

where $T_e$ and $T_d$ are the electromagnetic and load torques, respectively; $J_r$ and $B_f$ are the inertias of the rotor and friction coefficients, respectively; and $i_a$, $i_b$, and $i_c$ are the measured phase currents. The Clarke transformation was applied to them to determine the stator current in the stationary *d-q* frame. The Park transformation was applied to obtain this projection in a synchronous *d-q* frame with rotor position information. When the actual speed or load torque changed, the reference *d-q* currents immediately adjusted the speed and torque. Once the appropriate adjustment was accomplished, the motor speed became the reference speed, and the motor quickly achieved a steady-state operation. Owing to the current loop, the actual current tracked the commanded current.

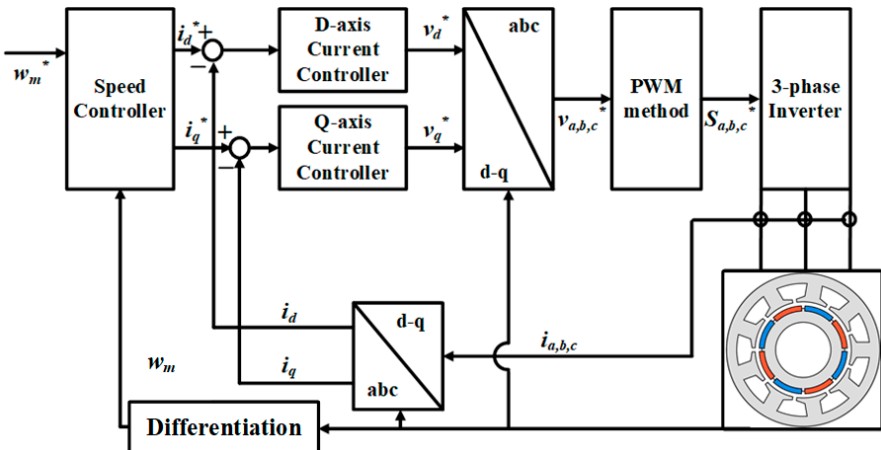

**Figure 2.** Vector control diagram of PMSM for dynamic analysis.

## 3. Dynamic Analysis of PMSM Based on Rotor Eccentricity

A no-load analysis must be performed for a dynamic analysis. The no-load analysis also has the purpose of verifying that the analysis model is well set up and is necessary to confirm that the voltage limit condition is satisfied in the control process. In addition, since back EMF is the element that primarily affects the control aspect, no-load analysis must be performed. Next, dynamic analysis was performed using the modeled controller, PWM method, and switching, and analysis was performed to store data for each location that performed co-simulation.

### 3.1. No-Load Analysis

The back EMF is directly affected by the rotor eccentricity and depends on the pole/slot combination. In this analysis, an eight-pole, nine-slot PMSM was selected to confirm the effect of rotor eccentricity. Figure 3 shows the magnetic flux density of the analyzed model. When comparing the case with no eccentricity to cases with 1 and 1.5 mm eccentricities, it was confirmed that the magnetic flux density formed more unevenly when eccentricity occurred. A no-load analysis was performed to analyze the back EMF of the analysis model based on the rotor eccentricity at 0, 1, and 1.5 mm, in which it was confirmed that additional harmonics were generated from the unbalanced back EMF. The harmonics of the unbalanced back EMF affected the control quality consequently. Figure 4 shows the back EMF results based on the rotor eccentricity, where "ec" denotes the eccentricity ratio. It represents the value of how eccentricity occurred based on the length of the nominal air gap. Figure 5 shows the results of fast Fourier transform (FFT) analysis of the back EMF. Figure 5a–c indicates ec = 0, 1, and 1.5 mm, respectively. Harmonics, except for the fundamental wave, are shown to compare the harmonics in

each case. When the eccentricity was 1 or 1.5 mm, additional harmonics were generated at the back EMF. Furthermore, the zeroth, second, and third harmonics increased significantly with the amount of eccentricity. The magnitude of the harmonics is summarized in Table 2. These harmonics adversely affected the dynamic characteristics of the PMSM, and the core loss increased.

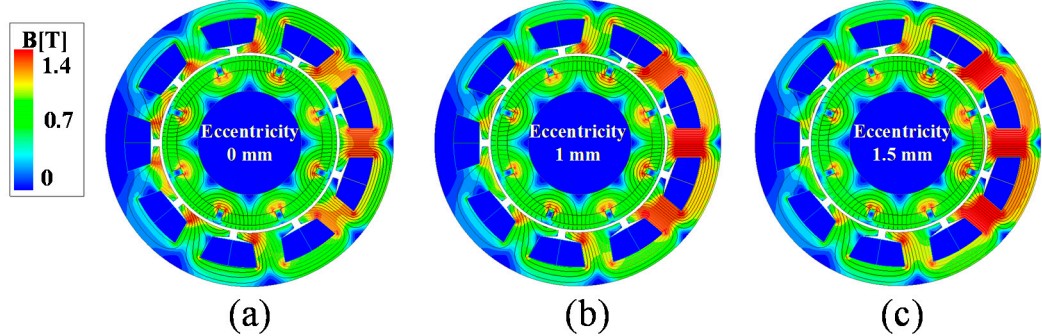

**Figure 3.** Magnetic flux density of analysis model: (**a**) ec = 0 mm, (**b**) ec = 1 mm, and (**c**) ec = 1.5 mm.

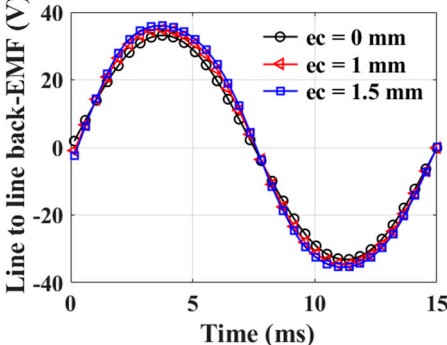

**Figure 4.** One-phase back electromotive force (EMF) based on rotor eccentricity ratio.

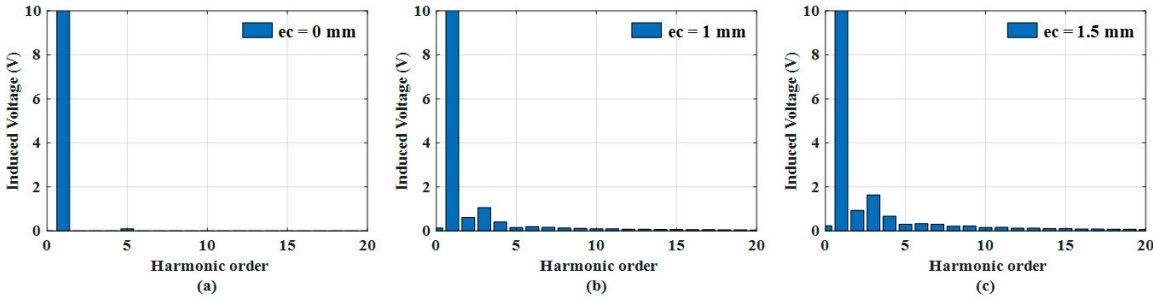

**Figure 5.** Fast Fourier transform analysis results of back EMF: (**a**) ec = 0 mm, (**b**) ec = 1 mm, and (**c**) ec = 1.5 mm.

**Table 2.** Fast Fourier transform analysis results of back EMF.

| Harmonic Order | ec = 0 mm (V) | ec = 1 mm (V) | ec = 1.5 mm (V) |
|---|---|---|---|
| 0 | 0 | 0.13 | 0.23 |
| 1 | 33.2 | 35.9 | 37.4 |
| 2 | 0 | 0.6 | 0.9 |
| 3 | 0 | 1.05 | 1.6 |
| 4 | 0 | 0.4 | 0.7 |
| 5 | 0.1 | 0.16 | 0.3 |
| 6 | 0 | 0.2 | 0.3 |
| 7 | 0 | 0.16 | 0.3 |
| 8 | 0 | 0.13 | 0.2 |

### 3.2. Co-Simulation Analysis

Co-simulation is one of the coupled analysis methods used to shorten the analysis time. By storing data such as torque and magnetic flux for each position of the rotor, the time required for dynamic analysis can be shortened, and almost the same results as in coupled analysis can be obtained. Mathematical models of controller and PWM inverter of the PMSM were modeled, and the dynamic analysis was conducted similarly to the coupled analysis.

### 3.3. Core Loss Analysis

The core loss can be calculated using the well-known Steinmetz equation as follows:

$$P_{core} = P_h + P_e + P_a = K_h f B_c^{n_{st}} + K_e f^2 B_c^2 + K_a f^{1.5} B_c^{1.5} \tag{7}$$

where $P_h$, $P_e$, and $P_a$ are the hysteresis loss, eddy current loss, and anomalous loss, respectively. Here, $K_h$, $K_e$, $K_a$, and $n^{st}$ are the hysteresis loss coefficient, eddy current loss coefficient, anomalous loss coefficient, and Steinmetz constant, respectively; and $f$ and $B_c$ are the frequency and the external magnetic flux density in the core. Hysteresis loss was yielded because energy was lost each time a hysteresis loop traversed. This loss is directly proportional to the size of the hysteresis loop of the used material and the amplitude of the excitation. The Steinmetz constant is a material-dependent exponent that is typically between 1.5 and 2.5 [13]. Table 3 shows the core loss coefficients of the core used based on frequency. The core loss data for each frequency was provided by the core manufacturer, and each coefficient can be derived by using the curve fitting method.

**Table 3.** Core loss coefficient based on curve fitting.

| $f$ (Hz) | $K_h$ (w/m³) | $K_e$ (w/m³) | $K_a$ (w/m³) | $n_{st}$ |
|---|---|---|---|---|
| 50 | 199.9 | 0.67 | 0 | 2 |
| 60 | 195 | 0.77 | 0 | 2 |
| 100 | 181.1 | 0.12 | 7.8 | 2 |
| 200 | 182.4 | 0.5 | 4.4 | 2 |
| 400 | 199.9 | 0.67 | 0 | 2 |

## 4. Results and Discussion

Figure 6 shows the experimental set, which was constructed to control the amount of rotor eccentricity. To measure the back EMF at the rated speed, a back-to-back system was constructed, wherein one servo motor and one test motor were arranged back to back. In addition, the rotor eccentricity was precisely controlled using two gap sensors. Table 1 lists the specifications of the motor. Figures 7 and 8 show the input current from the inverter to the PMSM based on the eccentricity. Figure 7a–c indicates ec = 0, 1, and 1.5 mm, respectively. Even when eccentricity occurred, the current magnitude was almost the same in each case; however, it was observed that additional harmonics were generated in each case. To compare the harmonics more precisely, the FFT results of each case were analyzed, except the zeroth to third harmonic components. In the experiment, the initial position was determined using an encoder, however it is considered not smooth due to the inverter switching frequency. Figure 9 shows the core loss when the fundamental current wave was applied, and Figure 10 shows the core loss when harmonics were considered in the applied current. The current used in the core loss analysis was derived from the experimental results. It was confirmed that when harmonics were considered in the applied current, the ripple and average value increased. Comparing the average value of each case, the values increased by approximately 12%, 11.5%, and 10.6%.

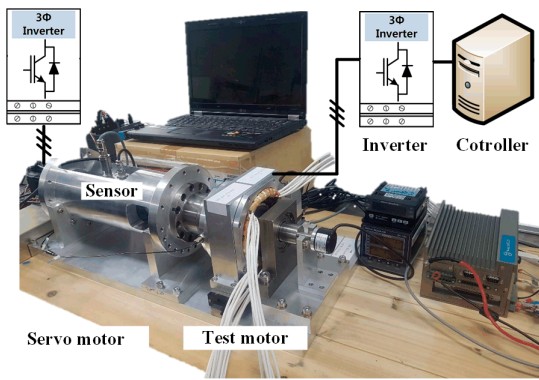

**Figure 6.** Experimental setup.

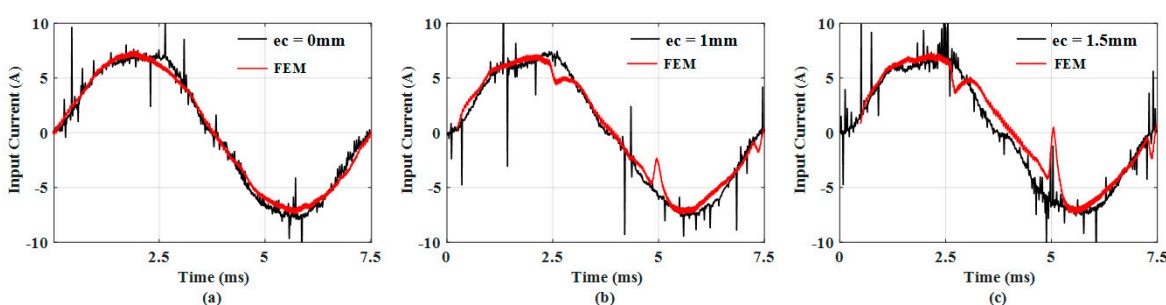

**Figure 7.** Experimental results of input current: (**a**) ec = 0 mm, (**b**) ec = 1 mm, and (**c**) ec = 1.5 mm.

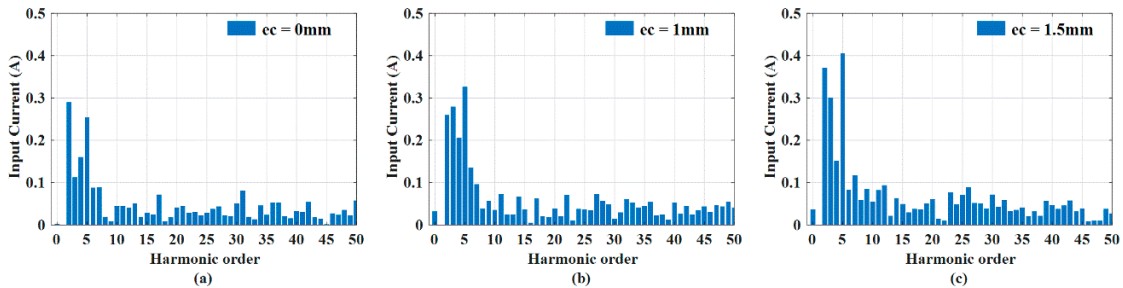

**Figure 8.** Experimental fast Fourier transform (FFT) results of input current: (**a**) ec = 0 mm, (**b**) ec = 1 mm, and (**c**) ec = 1.5 mm.

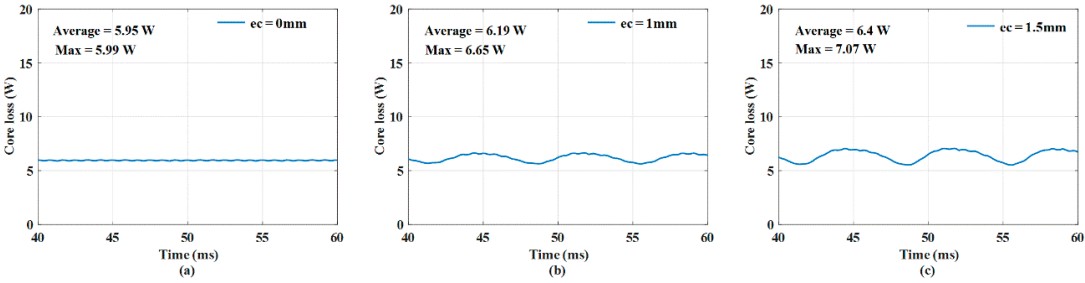

**Figure 9.** Core loss when fundamental current was applied: (**a**) ec = 0 mm, (**b**) ec = 1 mm, and (**c**) ec = 1.5 mm.

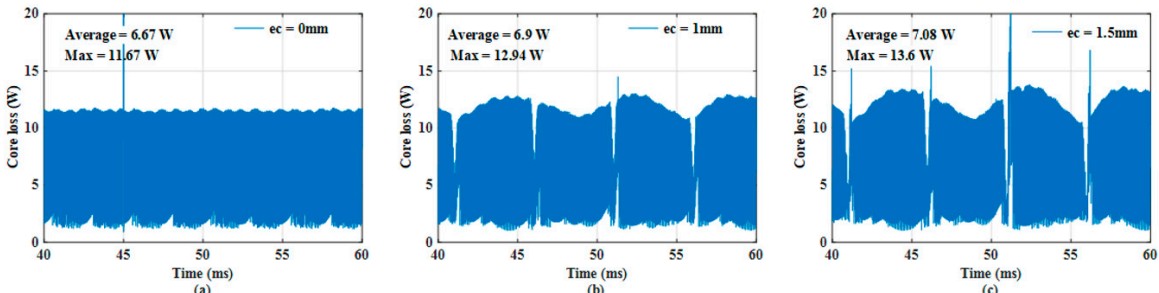

**Figure 10.** Core loss when harmonics were considered in applied current: (**a**) ec = 0 mm, (**b**) ec = 1 mm, and (**c**) ec = 1.5 mm.

## 5. Conclusions

In this study, we performed an electromagnetic characteristic analysis of a PMSM considering the current waveform based on rotor eccentricity. First, the PMSM was modeled mathematically. A no-load analysis and a co-simulation were performed to analyze the dynamic characteristics of the PMSM when rotor eccentricity occurred. We confirmed the additional harmonics of the input current supplied from the inverter to the motor in the presence of rotor eccentricity in each result. The modeling of the PMSM considering rotor eccentricity was performed using ANSYS Maxwell, and the dynamic analysis of the PMSM was performed using ANSYS Twin Builder. We confirmed that the current harmonics increased when rotor eccentricity occurred by comparing the analysis and experimental results. It was confirmed that a larger core loss occurred when the current harmonics were considered. These results are expected to facilitate the analysis of PMSM characteristics based on rotor eccentricity when the motor is driven.

**Author Contributions:** J.Y.C.: conceptualization, review and editing; H.K.L.: analysis, original draft preparation; T.K.B.: motor control algorithm; J.H.W.: co-simulation; H.S.S.: core loss analysis. All authors have read and agreed to the published version of the manuscript.

**Funding:** This work was supported by the Korea Institute of Energy Technology Evaluation and Planning (KETEP) and the Ministry of Trade, Industry & Energy (MOTIE) of the Republic of Korea. (No. 20183010025420) and This work was supported by the National Research Foundation of Korea(NRF) grant funded by the Korea government(MSIT). (No. 2020R1A2C1007353)

**Conflicts of Interest:** The authors declare no conflict of interest.

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
