# Peer review of "Electromagnetic Characteristic Analysis of Permanent Magnet Synchronous Machine Considering Current Waveform According to Static Rotor Eccentricity"

_applsci, doi:10.3390/app10238453_

Round 1
Reviewer 1 Report
I have put some remarks/corrections in the body of your paper attached to the review. These are:
- a) line 26 – change the position of the references
- b) line 36 – driving environment of the motor – I suggest to change it to “the whole drive system”
- c) line 92, Fig.2 – Differentiation
- d) line 105 – Clark trans. is commonly described by “α-β”
- e) lines 107-110 – I suggest not to use passive e.g. replace sentences like “When the actual speed or load torque changed” with “When the actual speed or load torque changes” and so forth,
- d) line 112 – the
- e) line 121 – replace “analysis model” with “analyzed model”
- f) line 132 – you mentioned the existence of 0th (constant) component in the phase voltage of BackEMF. But usually the phase voltage is not available for measurement as a star without neutral wire is a standard winding connection scheme. Could you make a similar analysis for line-to-line voltage?. I would also like you to add a table with values of harmonic components for Fig.5 to include the 1st harmonic for comparison,
- g) line 148 – “as in”
- h) line 149 – models ; created
- i) line 159 – and
- j) line 179-180 – what do you mean?

Author Response
Dear reviewers:
Manuscript ID: applisci-996700
Title: Electromagnetic Characteristic Analysis of Permanent Magnet Synchronous Motor Considering Current Waveform according to Rotor Eccentricity
Authors: Hoon-Ki Lee, Tae-Kyoung Bang, Jong-Hyen Woo, Hyo-Seob Shin, Jang-Young Choi.
We are sending a cover letter and revised manuscript.
Please see the attachment.
Thanks for your kindness.

Reviewer 2 Report
This is well written and interesting paper. The style of presentation is quite clear but in my opinion this paper is a good engineering work but it is not an original contribution to the scientific community. The work proposed in this paper is not new and original and I do not see how this paper adds anything novel or even incremental to the existing literature.
Presently a lot of research works are focuses on an impact of rotor eccentricity on dynamic characteristics in the PMSMs. They addresses studies on topics such as: rotor eccentricity effect on the vibration and noise and/or on the PMSM transients characteristics in 2D or 3D. In a lot of these papers authors uses an in-house/author’s mathematical model of considered motors. I would like to emphasize that I do not dare to say that the developed numerical models by using the professional/commercial programs like Ansysy/Infolytica/Comsol/etc. cannot be used to analyze coupled phenomena in electrical machines, but these models should be measurement verified and also detailed described.
From the point of view of the quality of the research work, the authors should present in the paper more details:
- Abstract should be rebuild because in my point of view in its current form doesn't give a pertinent overview of the paper in detailed.
- Autors written: “Figure 3 shows the magnetic flux density of the analysis model. When comparing the case with no eccentricity and cases with 1 and 1.5 mm eccentricities, it was confirmed that the magnetic flux density formed more unevenly when eccentricity occurred.” In my opinion in Fig. 3a the distribution of the magnetic flux density is also unsymmetrical. Perhaps the authors should be consider present the lines distribution of the magnetic flux.
- What kind of eccentricity is analyzed in the paper? Static, dynamic, "mixed"?
- Data of the motors (supplied voltage, input current, nominal torque, core lossess) should be presented.
- Mathematical model of PMSM is well known and doesn't add anything novel.
- Line 104: …ias and ibs symbols are not considered in the paper.
- Line 112: „A no-load analysis must be performed for a dynamic analysis and to verify The no-load analysis…” the dot is missing.
- Line 137/144: Legend of the Figure 3 and 5 should be completed by “…and (c) ec = 1.5 mm.”
- Equation (8) enables to calculate the core loss in the frequency domain in Maxwell software. In accordance with my experiences we can only determine the eddy current loss factor using analytical methods. The analytical methods cannot calculate the other two factors. Please explain how did you calculate the others factors?
- Figure 7 shows the experimental results of input current obtained on the basis of the test setup including inverter to supply PM. Please explain why the current waveforms are not smooth. In my opinion the location of the inverter in relation to the measurement system affects the shape of the current and, consequently, its harmonics.
- Applied solver and assumed level of accuracy for carried out FEM simulations e.i. details of the mesh in FEM model in Fig.1a (e.g. number of elements, nodes, time consumption of simulation, time-step length).
- The experimental results are not compared to the results of the FE model which makes it harder to judge the accuracy of the numerical model elaborated in ANSYS Maxwell.
Summary:
From my point of view, the paper cannot be published in its current form.
Author Response

(The authors gave the same response as above.)

Round 2
Reviewer 2 Report
Thank you for your answers. I have no more comments.